# Measurement report: Ice nucleating particles active ≥ -15 °C in free tropospheric air over western Europe

Franz Conen[1], Annika Einbock[1], Claudia Mignani[1], Christoph Hüglin[2]

[1]Department of Environmental Sciences, University of Basel, CH-4056 Basel, Switzerland

[2]Laboratory for Air Pollution / Environmental Technology, Empa, CH-8600 Dübendorf, Switzerland

*Correspondence to*: Franz Conen (franz.conen@unibas.ch)

**Abstract.** Ice nucleating particles (INP) initiate ice formation in supercooled clouds, typically starting in western Europe at a few km above ground. However, little is known about the concentration and composition of INP in the lower free troposphere (FT). Here, we analysed INP active at -10 °C (INP$_{-10}$) and -15 °C (INP$_{-15}$) collected during FT conditions at the high-altitude

observatory Jungfraujoch between January 2019 and March 2021. We relied on continuous radon measurements to distinguish FT conditions from those influenced by the planetary boundary layer. Median concentrations in the FT were 2.4 INP$_{-10}$ m$^{-3}$ and 9.8 INP$_{-15}$ m$^{-3}$, with a multiplicative standard deviation of 2.0 and 1.6, respectively. A majority of INP was deactivated after exposure to 60 °C, thus probably originated from certain epiphytic bacteria or fungi. Subsequent heating to 95 °C deactivated another 15% to 20% of the initial INP, likely other types of fungal INP that might be associated with soil organic

matter or with decaying leaves. Very few INP$_{-10}$ withstood heating to 95 °C, but on average 20% of INP$_{-15}$ in FT samples did so. This percentage doubled during Saharan dust intrusions, which had practically no influence on INP$_{-10}$. Overall, the results suggest that aerosolised epiphytic microorganisms, or parts thereof, are responsible for the majority of primary ice formation in moderately supercooled clouds above western Europe.

## 1 Introduction

Free troposphere (FT) designates a part of the troposphere that only occasionally is in exchange with Earth's surface, whereas the planetary boundary layer (PBL) continuously exchanges particles with surface sources and sinks. Consequently, the FT integrates particle emissions over a much larger area than does the PBL. A community of airborne microorganisms, for example, is in the PBL mainly composed of organisms emitted from sources within a distance of several tens of kilometers around an observation point (Tignat-Perrier et al., 2019). In contrast, particle populations sampled at a high-altitude mountain

station under FT conditions constitute a mixture of many sources and sinks active on a continental scale (Herrmann et al., 2015, Sun et al., 2021). A special kind of aerosol particles, so-called ice-nucleating particles (INP), is relevant for primary ice formation in mid-level clouds (Findeisen, 1938). Ice formation in clouds above lowlands in western Europe starts at a few kilometers altitude at around -5 °C (Kanitz et al., 2011). However, FT air masses have been little explored in terms of the INP they carry, let alone INP active > - 15 °C. For recent summaries of INP studies in the FT see Lacher et al. (2018) and He et al.

(2021). In FT conditions at the high-altitude observatory Jungfraujoch (3580 m a.s.l.) in the Swiss Alps, Lacher et al. (2018) found similar concentrations of INP active at -31 °C (INP$_{-31}$) as had been reported for the FT in other regions of the world (summarised in Table 2 and Figure 6 in Lacher et al., 2018), and little seasonal variation. Later, continuous measurements by Brunner et al. (2021b) found the monthly median of INP$_{-30}$ in the FT background to peak in April and to be lowest in December. Brunner et al. (2021a, 2021b) found most INP$_{-30}$ at Jungfraujoch occur during Saharan dust intrusions, by far exceeding any

background concentration. Yet, in the PBL and at more moderate supercooling (here: ≥ -15 °C), biological particles seem to constitute the majority of the INP population as revealed by heat tests (Hill et al., 2016). Heat deactivates biological INP but leaves mineral INP largely unaffected (Hill et al., 2016). Chen et al. (2021) found during dust events sampled in Beijing that 70% of INP active at ≥ -15 °C (INP$_{-15}$) were heat-sensitive. Observed heat-sensitive fractions of INP$_{-15}$ in the PBL of agricultural areas in the USA (Suski et al., 2018) and in Argentina (Testa et al., 2021) were even larger (> 90%). These findings

contrast with a small biological fraction detected among ice particle residuals collected at Jungfraujoch from mixed-phase clouds and classified by physico-chemical analyses (Mertes et al., 2007) or by laser ablation mass-spectrometry (Schmidt et al., 2017; Lacher et al., 2021). Not every ice particle residual has initiated as INP the formation of the ice particle it was recovered from, in particular not when it was recovered from a secondary ice particle. An ice particle residual recovered from a primary ice particle and classified by mass spectrometry as mineral dust may have been activated at moderate supercooling by a minor, ice-nucleation active biological component sticking to its surface (Augustin-Bauditz et al., 2016). A heat test, which specifically targets the ice-nucleation active component, would have classified the same assembly as biological INP.

Aerosolisation of biological particles from vegetated land to the PBL is intensified during rainfall (Huffman et al. 2013; Prenni et al., 2013; Iwata et al., 2019). Subsequent transport from the PBL to the FT may happen through mixing in frontal systems, cloud convection, or by mountain venting (Henne et al., 2005). Here, we investigate whether biological INP, as classified by heat tests, also contribute the majority of the INP population in the FT. Our objectives were to quantify INP active at moderate supercooling in the FT, to narrow down the likely composition of these INP and how it might change during Saharan dust (SD) intrusions. We will also compare INP concentrations in the FT with those of air masses influenced by the PBL as presented in earlier studies.

## 2 Material and methods

### 2.1 Sampling and analysis

For our investigation we relied on archived samples of particulate matter ($PM_{10}$), collected between January 2019 and March 2021 (Table A1) on quartz-fibre filters at the high-altitude observatory Jungfraujoch (07°59'02" E, 46°32'53" N, 3580 m.a.s.l.), which is part of the Swiss national air quality monitoring network. At the site, $PM_{10}$ sampling and INP analysis have been described earlier (Conen et al., 2012). Units of air volume throughout this paper are normalised to standard pressure. Briefly, air was sampled at a rate of 720 m$^3$ day$^{-1}$ through a heated inlet, followed by a $PM_{10}$ cut-off, onto a 150 mm diameter quartz-fibre filter (Pallflex Tissuquartz 2500-UP) with 140 mm diameter of active area. Every day at midnight a new filter was exposed to the continuous air stream for 24 hours. From each selected filter we took 72 punches with 2 mm diameter each; all 72 punches together contained aerosol particles from a total of 10.6 m$^{-3}$. Each punch was placed into a 0.5 mL Eppendorf safe-lock tube and 0.1 mL ultra-pure water (Sigma-Aldrich, W4502-1L) was added. The tubes were then immersed in a cooling bath and exposed to a temperature ramp from -5 °C to -15 °C (0.3 °C min$^{-1}$). After each 1 °C step the number of frozen tubes was counted by eye and the cumulative concentration of INP was calculated according to Vali (1971). After the first such freezing assay, the tubes were placed for 10 min in a water bath at 60 °C before the freezing assay was repeated. It was followed by a second 10-min heat treatment at 95 °C and a third freezing assay. Loss of INP active < -10 °C due to the repeated freezing is unlikely in this study. Although Polen et al. (2016) have observed throughout five repeated freezing assays some loss of INP active > -5 °C, little had changed in INP active < -5 °C (see Fig. 6 in Polen et al., 2016). A more extensive set of tests was conducted by Vali (2008) on a soil sample. After 55 refreezing cycles an increasingly larger fraction of INP had been lost above -10 °C toward the warmer end of the freezing spectrum, but the concentration of INP active < -10 °C had remained practically the same (Fig. 1c in Vali, 2018).

Probably the largest uncertainty in these assays is related to an often small number of INP assessed. The 68% confidence interval (± 1 σ) of a determined INP number concentration is roughly proportional to the square root of INP in an assay. In this study, an assay contained particles collected from 10.6 m$^3$ of air. Hence, a concentration of 1 INP m$^{-3}$ was represented by a total of 10 or 11 INP in the assay and is associated with an uncertainty of around ± 0.3 INP m$^{-3}$ (√11 / 11 x 1 INP m$^{-3}$). Two blank (background) assays were done with punches from 5 mm wide fringes of sample filters. This part of a filter is covered

by the clamp rings holding it in place during sampling. That way, sampled air does not pass through it and it remains clean. However, during handling and transport some particles may get smeared from the active filter area onto this narrow fringe. Consequently, these blank values are a conservative (upper) estimate of a field blank. Each of these blanks was composed of punches from four filters. Sample values were corrected for blank values by subtracting the average of both blanks, which on average was 7% of a sample value. We calculated by difference the number concentration of heat sensitive INP (INP active before any heat treatment - INP active after 60 °C treatment) and moderately heat tolerant INP (INP active after 60 °C treatment - INP active after 95 °C treatment). What remained active after the 95°C treatment is termed heat tolerant. In one sample (06.02.2021), the assay was completely frozen at -15 °C and was repeated with smaller punches (1 mm diameter).

## 2.2 Identification of free tropospheric condition

A key issue in our study is the distinction between FT and PBL-influenced air masses. For this purpose, we routinely monitor radon ($^{222}$Rn) at Jungfraujoch with a dual-loop two-filter detector (Whittlestone and Zahorowski, 1998). Radon is emitted relatively homogenously in space and time from land surfaces. Its only sink in the atmosphere is radioactive decay. Because of its short half-life (3.82 days), the concentration difference is large between the FT and the PBL, making radon a useful tracer to discriminate between these two types of air masses (Herrmann et al., 2015; Chambers et al., 2016). The probability density function of all hourly mean radon concentrations measured over the past five years is well reproduced by the sum of two log-normal distributions (Fig. 1), most likely representing FT and PBL-influenced air masses (Conen et al., 2021, Brunner et al., 2021a). Although FT conditions prevailed 41.5% of the time, it was only 1 in 13 days in 2019 and 2020 that such conditions applied to every hour of a full 24-hour PM$_{10}$ sampling period. Here, we consider a 24-hour PM$_{10}$ sampling period as representative of the FT when all 24 values of hourly mean radon concentration during the sampling period were below 1.05 Bq m$^{-3}$. Ninety percent of all values below 1.05 Bq m$^{-3}$ belong to the distribution representing FT conditions (Fig. 1). Two-thirds of the FT samples defined that way were collected in the months of January, February, and March. We selected for INP analysis 28 such samples from between January 2019 and March 2021, making sure to have samples from each month of the year for which they were available. In addition, we selected from the filter archive an additional seven samples with high PM$_{10}$ loads and a yellow-brown (ochre) colour resulting from SD intrusions (Collaud Coen et al., 2004) .

The median radon concentration in the FT distribution is only 0.28 times that of the PBL-influenced distribution (Fig. 1), hence, PBL injections in the FT may have a median age of about seven days. Thus, the airshed from where INP in FT air masses sampled at Jungfraujoch might have originated, is roughly indicated by seven-day backward trajectories calculated by the HYSPLIT model (Fig. 2). One trajectory was calculated for each day, ending at 13:00 UTC at 500 m above ground level, which for Jungfraujoch is 1500 m above sea level in the model surface. The model was run with the following settings: meteorology: GDAS (1 degree, global), vertical motion calculation method: model vertical velocity, location coordinates: 46°32'53'' N, 7°59'2'' E.

## 3 Results and discussion

### 3.1 Concentration and likely composition of ice nucleating particles in free tropospheric air

The observed range of INP concentration in FT air masses was 1.0 m$^{-3}$ to 5.6 m$^{-3}$ for INP active at -10 °C (INP$_{-10}$) and 4.1 m$^{-3}$ to 16.3 m$^{-3}$ for INP active at -15 °C (INP$_{-15}$). These values are within the lower range of INP concentrations observed in other contexts (Petters and Wright, 2015). The INP sampled at Jungfraujoch during FT conditions likely originated from the northern part of western Europe and from the North Atlantic (Fig. 2). Latter influence was probably minor. McCluskey et al. (2018) found a mean of 1.1 INP$_{-15}$ m$^{-3}$ in pristine marine air masses arriving at the west coast of Ireland, one tenth of the median we found at Jungfraujoch. A majority of INP$_{-10}$ (83%) and INP$_{-15}$ (57%) in the FT were heat sensitive and lost their activity after

exposure to 60 °C (Table 1; for INP active at other temperatures between -8 °C and -15 °C see Table A3). Possible contributors to this category include the bacteria *Erwinia herbicola* (Phelps et al., 1986) and *Pseudomonas sp.* (Pouleur et al., 1992), as well as heat sensitive spores of several fungal species (Table 2). These microorganisms live on plant surfaces from where they, or parts of them, are emitted to the atmosphere (Lindow et al., 1978; Lindemann et al., 1982; Hirno and Upper, 2000; Huffman et al., 2013). Further potential sources include other fungi, such as *Fusarium graminearum* (Vujanovic et al., 2012; Keller et al., 2014) and *Puccinia sp.* (Morris et al., 2013). However, we do not know whether these already lose their ice nucleation activity at 60 °C, or between 60 °C and 95 °C, as heat stress tests were done only at around 95 °C, which is why they are not included in Table 2.

Between 15% and 20% of INP we found in the FT were moderately heat tolerant (deactivated between 60 °C and 95 °C). This deactivation temperature matches the profile of *Mortierella alpina*, a saprophytic fungus associated with decaying leaf litter (Vasebi et al., 2019) and with soil particles (Fröhlich-Nowoisky et al., 2015; Conen and Yakutin, 2018). Other potential sources of moderately heat tolerant INP include fungal symbionts in lichen (Kieft, 1988), *Fusarium avenaceum* (Pouleur et al., 1992), and the other above-mentioned fungi of which the deactivation temperature is not clearly defined. The sum of heat sensitive and moderately heat sensitive $INP_{-15}$ in a FT sample was on average 80% (standard deviation = 9%) of its total $INP_{-15}$ content, little smaller than the combined fractions of fungal and bacterial $INP_{-15}$ (92%) estimated in a case study over Amazonia (Patade et al., 2021).

Heat tolerant INP (not deactivated at 95 °C) constituted a negligibly small fraction of all $INP_{-10}$, but to $INP_{-15}$ they contributed as much as did the moderately heat tolerant ones. The fraction of heat tolerant $INP_{-15}$ should substantially increase during SD intrusions, if mineral dust, in particular K-feldspar, was their main component (Vergara-Temprado et al., 2017), an issue addressed in the next section.

**3.2 Effect of Saharan dust intrusions**

Mineral dust makes by far the largest contribution to INP activated at around -30 °C at Jungfraujoch (Larcher et al., 2018, 2021; Brunner et al., 2021a, 2021b). Most INP in airborne mineral dust are probably K-feldspars, some of which are already activated at temperatures above -10 °C (Boose et al., 2016; Harrison et al., 2019). Here, we analysed seven samples collected during SD intrusions (Annex, Table A1), but excluded one sample from further analysis (06.02.2021), in which INP had substantially increased at -15 °C after each heat treatment. A similar reaction to heating was previously observed by Boose et al. (2019) and explained by the removal of secondary organic coatings that had masked ice-nucleation active sites before the heat treatment. In the remaining six samples, median $PM_{10}$ concentration was 19 times as large as in FT samples. Surprisingly, the median concentration of heat tolerant $INP_{-15}$ during SD intrusions was larger by only a factor of four, as compared to the FT (Table 1). Hence, mineral dust cannot be the main heat tolerant $INP_{-15}$ in the FT. Otherwise, SD intrusions, which consist mainly of mineral dust, should have led to an increase in the heat tolerant fraction of $INP_{-15}$ that is roughly proportional to the increase in dust load. This assumption is based on the finding that a large fraction of particle volume (i.e. mass) during a SD event at Jungfraujoch falls into the size range of particles > 0.5 μm in optical diameter (Schwikowski et al., 1995), which is a good predictor of atmospheric $INP_{-15}$ (DeMott et al., 2010; Mignani et al., 2021). Since we found no peak in heat tolerant $INP_{-15}$ during the main pollen season in spring, pollen and pollen fragments (Diehl et al., 2002; Pummer et al., 2012) are also an unlikely source of them (note: recently, Burkart et al. (2021) found convincing evidence that ice nucleation activity in birch pollen is related to a protein. Therefore, it probably is also heat sensitive, contrarily to earlier conclusions). More likely candidates of the heat tolerant fraction in the FT include soil organic material stabilised through bonding to minerals surfaces against deactivation by heat (Perkins et al., 2020), or lignin, not necessarily bound to mineral surfaces (Bogler and Borduas-

Dedekind, 2020). Such organic compounds are more abundant in fertile soils in the mid-latitudes as compared to desert soils and could explain the comparatively small effect of SD intrusions on heat tolerant INP$_{-15}$ observed at Jungfraujoch.

Further, the median of heat sensitive and of moderately heat tolerant INP$_{-15}$ was larger during SD intrusions by a factor of 1.6 and 2.3, respectively (Table 1). Their increase may be due to a concurrent influence of the PBL with all SD intrusions, as suggested by enhanced radon concentrations (Annex, Table A1). Overall, intrusions of SD and the associated PBL influence had doubled the median concentration of INP$_{-15}$ and doubled the average fraction of heat tolerant INP$_{-15}$ in a sample, but had no discernable effect on the median concentration of INP$_{-10}$ (Table 1).

**3.3 Comparison with other studies in the Swiss Alps**

Mignani et al. (2021) have made a large number of impinger-based INP$_{-15}$ measurements (n = 124, each integrating over 20 min) in February and March 2019 at Weissfluhjoch, a mountain station 145 km east of Jungfraujoch, and at a 909 m lower elevation (2671 m a.s.l.). This station was throughout the day increasingly influenced by air from the PBL (Wieder et al., 2021). The median of all observations, excluding those influenced by SD (n = 11), was 15.5 m$^{-3}$ (s* = 4.0 m$^{-3}$). This value is

about a factor of two larger than the median of the four FT samples collected at Jungfraujoch during the same months, February and March 2019 (8.9 m$^{-3}$, range: 5.5 m$^{-3}$ to 13.0 m$^{-3}$). A factor of two decrease with an increase of 1000 m in altitude was already observed for INP$_{-8}$ in Switzerland (Conen et al., 2017). Similar to the present study, SD intrusions at Weissfluhjoch enhanced the median INP$_{-15}$ concentration by much less than the concurrent increase in particle number concentration.

A surprising feature of INP$_{-10}$ in FT samples is their narrow distribution (1.0 to 5.6 m$^{-3}$) throughout the year (Fig. 3), which is in contrast to what we had found earlier at Jungfraujoch in filters not selected for FT conditions that covered a range of three orders of magnitude (Conen et al., 2015). Those filters were sampled between June 2012 and May 2013 and analysed by the same method as used in this study, but without the heat treatments. Generally, the influence of the PBL on Jungfraujoch is largest in summer and smallest in winter (Collaud Coen et al., 2011; Griffith et al., 2014). Hence, if samples are not explicitly

selected for FT conditions, the majority of them will have at least some PBL influence, in particular during summer (Brunner et al., 2021b). Therefore, concentration of INP$_{-10}$ on filters not selected for FT conditions occasionally exceeded that of filters selected for FT conditions, in particular during summer (Fig. 3). However, the opposite was true for winter. Especially during January and February, the concentration of INP$_{-10}$ was smaller by one or two orders of magnitude in the randomly selected samples of the year 2013, as compared to the FT samples of the years 2019, 2020, and 2021. A plausible explanation for this

difference could be that in 2013 a substantially larger fraction of INP$_{-10}$ had been activated and deposited before reaching Jungfraujoch, as compared to the later years. Indeed, mean air temperature during sampling in January and February 2013 at Jungfraujoch was -16.2 °C (s.d. 5.3 °C), which is 6.4 °C colder than during sampling in January and February 2019, 2020, and 2021 (-9.8 °C; s.d. 4.6 °C). Yet, high relative humidity and low temperature does not always result in a particularly low number concentration of INP$_{-10}$ recorded at Jungfraujoch. On 14.03.2021 an exceptionally strong storm from North-West passed

Jungfraujoch with gusts up to 122 km h$^{-1}$ and a mean daily windspeed of 51.5 km h$^{-1}$. Daily mean values of relative humidity and temperature were 98% and -18.8 °C, respectively. Nevertheless, we found a non-negligible concentration of INP$_{-10}$ (0.7 m$^{-3}$) on the PM$_{10}$ filter from that day. Updraft velocity at Jungfraujoch can reach a six-minute mean value of 8 m s$^{-1}$ during north-westerly wind (Hammer et al., 2014). An INP$_{-10}$ activated 1350 m below Jungfraujoch at -10 °C would grow for about 3 min ((18.8 °C - 10.0 °C) / 6.5 °C km$^{-1}$ / 8 m s$^{-1}$ = 169 s) into a crystal with a fall velocity of 5 cm s$^{-1}$ (Fukuta and Takahashi,

1999). As such, it can still enter the heated inlet of the sampler, which takes in droplets up to a size of about 40 μm (Weingartner et al., 1999), evaportate, and let the INP$_{-10}$ pass the PM$_{10}$ cut-off to be collected on the filter. At lower windspeed, growth time increases and activated INP$_{-10}$ are more likely to either settle or become too large to enter the heated inlet, eventually leading

to the very low concentrations, which might explain our earlier observations in 2013. In an increasingly warmer climate, seasonality in the sink term of INP active at moderately supercooled conditions above the Alps may fade.


In summary, the results of this study suggest that epiphytic microorganisms contribute the majority of INP to ice formation in moderately supercooled clouds above the northern part of western Europe, whereas the impact of Saharan dust is negligible at -10 °C and still limited at -15 °C.

**Acknowledgements**

We thank the International Foundation High Altitude Research Stations Jungfraujoch and Gornergrat (HFSJG), 3012 Bern, Switzerland, for making it possible for us to work and to operate instruments at the High-Altitude Research Station at Jungfraujoch. We thank Claudia Zellweger and Stefan Reimann at Empa for helping us with the selection of $PM_{10}$ filter samples and for generously sending sections of them to Basel. We are grateful to Alastair Williams and his group at ANSTO for the ongoing collaboration as the supplier and supporter of the radon detection system. The radon observations are supported

by the Swiss National Science Foundation (SNSF) as a contribution to the pan-European Integrated Carbon Observation System (ICOS) (https://www.icos-ri.eu). MeteoSwiss is acknowledged for the uncomplicated provision of meteorological data through its data portal IDAweb, and the Air Resources Laboratory at NOAA for providing access to its HYSPLIT model.

**Author contribution**

FC and CM conceived the study. CH organised the particle collection and provided the samples, AE did the freezing assays.
FC prepared the manuscript with contribution from all coauthors.

**Competing interests**

The authors declare that they have no conflict of interest.

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

**Table 1: INP concentration found in the free troposphere (FT) and during Saharan dust intrusions (SD), categorised according to heat sensitivity. Median and multiplicative standard deviation are shown. Use of the multiplicative standard deviation (s\*) is appropriate because multiplicative processes determine the concentration of biological particles in the atmosphere (Limpert et al., 2001, 2008). It was estimated from the upper (q3) and lower (q1) quartiles of a distribution: s\* = (q3/q1)0.741 (Limpert et al., 2001). It was not estimated where the lower quartile was below detection limit (< 0.095 INP m$^{-3}$). Heat sensitive are designated INP that**
**had lost their activity after exposure to 60 °C. Moderately heat tolerant are termed those INP that had retained their activity after exposure to 60 °C, but had lost it after exposure to 95 °C. Heat tolerant INP are those that were still active after both heat treatments.**

| Category | $INP_{-10}$ (m$^{-3}$) | | $INP_{-15}$ (m$^{-3}$) | |
|---|---|---|---|---|
| | FT (n = 28) | SD (n = 6) | FT (n = 28) | SD (n = 6) |
| all | 2.4 $_{2.0}$ | 2.4 $_{1.5}$ | 9.8 $_{1.6}$ | 20.0 $_{1.7}$ |
| heat sensitive | 2.0 $_{1.8}$ | 1.9 $_{1.6}$ | 5.6 $_{1.5}$ | 9.2 $_{2.1}$ |
| moderately heat tolerant | 0.4 $_{1.9}$ | 0.2 $_{n.d.}$ | 1.5 $_{2.8}$ | 3.5 $_{2.8}$ |
| heat tolerant | 0.0 $_{n.d.}$ | 0.1 $_{2.8}$ | 1.6 $_{1.6}$ | 6.2 $_{1.5}$ |

Table 2: Reported heat sensitivity of INP active at moderate supercooling. Indicated is the temperature range in which at least 90% of a specific type of INP active ≥ 15 °C was found to be deactivated, although a smaller fraction of the same INP may already have been deactivated at a lower temperature. For example, soil particles analysed by Conen and Yakutin (2018) had lost half of their INP-10 after exposure to 60 °C, but more than 98% after exposure to 95 °C. These particles are assigned a deactivation temperature between 60 °C and 95 °C. Note: studies where INP were found to be deactivated after a single heat treatment ~ 95 °C are not listed because deactivation might already have happened < 60 °C.

| Type | Deactivation temperature | | | Reference |
| --- | --- | --- | --- | --- |
| | < 60 °C | 60 °C - 95 °C | > 95 °C | |
| Bacteria | *Erwinia herbicola* | | | Phelps et al. (1986) |
| | *Pseudomonas sp.* | | | Pouleur et al. (1992) |
| | | | *Lysinibacillus sp.* | Failor et al. (2017) |
| Fungi | *Acremonnium implicatum* | | | Pummer et al. (2015) |
| | *Isaria farinosa* | | | Pummer et al. (2015) |
| | *Fusarium acuminatum* | | | Kunert et al. (2019) |
| | *Fusarium langsethiae* | | | Kunert et al. (2019) |
| | *Fusarium armeniacum* | | | Kunert et al. (2019) |
| | | Fungal symbiont in lichen | | Kieft (1988) |
| | | *Mortierella alpina* | | Fröhlich-Nowoisky et al. (2015) |
| | | *Fusarium avenaceum* | | Pouleur et al. (1992) |
| Others | | | | |
| mixed | | Leaf-derived ice nuclei | | Schnell and Vali (1973) |
| | | Soil particles | | Hill et al. (2016) |
| | | Soil particles | | Conen and Yakutin (2018) |
| organic | | | Pollen | Diehl et al. (2002), Pummer et al. (2012) |
| | | | Lignin | Bogler and Borduas-Dedekind (2020) |
| mineral | | | K-feldspars | Daily et al. (2021) |
| | | | Montmorillonite | Conen et al. (2011) |
| | | | Illite | O'Sullivan et al. (2015) |
| | | | Kaolinite | Daily et al. (2021) |


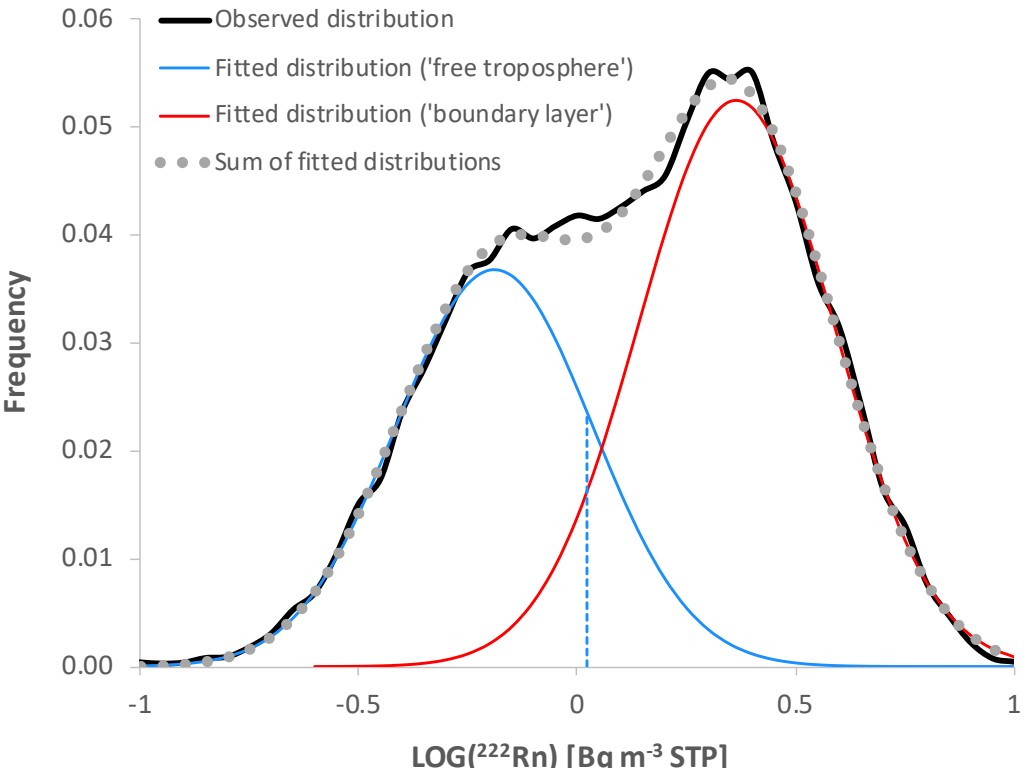

**Figure 1: Probability density function of log-transformed hourly mean radon concentration observed during the years 2016 to 2020 at Jungfraujoch (black line). The sum of two fitted and weighed log-normal distributions (grey dotted line) closely matches the observed distribution. The fitted distributions most likely represent the distribution of free tropospheric air masses (blue continuous line) and that of air masses influenced by the planetary boundary layer (red line). Ninety percent of the values below the dashed vertical line belong to the fitted 'free troposphere' distribution (Figure adapted from Conen et al., 2021).**




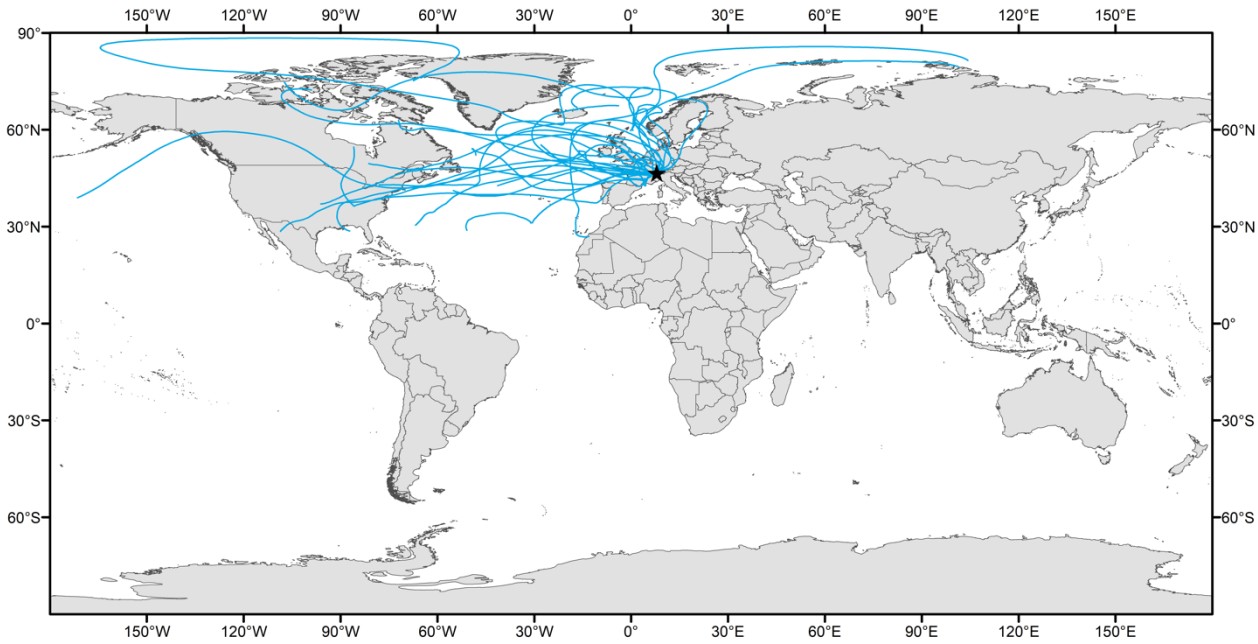


**Figure 2. Ensemble of seven-day backward trajectories ending at Jungfraujoch at noon local time on days for which we analysed INP in free tropospheric conditions. The trajectories were calculated by the HYSPLIT model, made available by the Air Resources Laboratory at NOAA (https://www.ready.noaa.gov/HYSPLIT.php).**


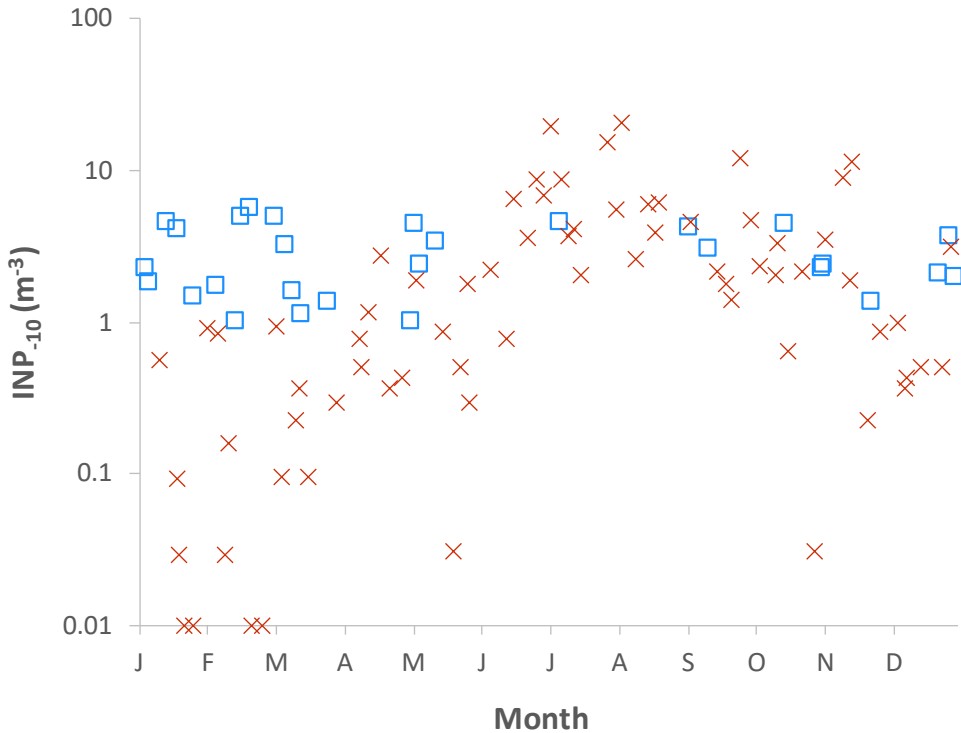

**Figure 3. Concentration of INP$_{-10}$ when free tropospheric conditions prevailed at Jungfraujoch thoughout a full day (open squares) and for days not selected for this criterion (crosses; data from Conen et al., 2015, see Annex, Table A2).**




**Table A1: Number concentration of ice nucleating particles active at -10 °C (INP$_{-10}$) and at -15 °C (INP$_{-15}$) at Jungfraujoch, determined for free tropospheric conditions and during Saharan dust intrusions. The same sample material was analysed 3 times. A first freezing assay was followed by a heat treatment (10 min at 60 °C) and second freezing assay, followed by a second heat treatment (10 min at 95 °C) and a third freezing assay. All steps were carried out in an uninterrupted operation. Blank values were subtracted from raw measured values. Three categories of INP were derived by difference between INP number concentration before and after each heat treatment, resulting in negative values where INP concentration had increased after a heat treatment. Meteorological data is from MeteoSwiss and PM$_{10}$ data from the Swiss air quality monitoring network (NABEL).**

| Date | INP$_{-10}$ (m$^{-3}$) | | | INP$_{-15}$ (m$^{-3}$) | | | T$_{mean}$ | Fraction of hours with RH > 95% | Radon | | | PM$_{10}$ |
|---|---|---|---|---|---|---|---|---|---|---|---|---|
| | | | | | | | | | min | max | mean | |
| | Deactivation temperature (°C) | | | Deactivation temperature (°C) | | | | | | | | |
| | < 60 | 60- 95 | > 95 | < 60 | 60- 95 | > 95 | (°C) | | (Bq m$^{-3}$) | | | (µg m$^{-3}$) |
| *Free tropospheric conditions* | | | | | | | | | | | | |
| 06.01.19 | 1.6 | 0.3 | 0.0 | 5.1 | 0.7 | 1.1 | -13.1 | 0.00 | 0.32 | 0.48 | 0.39 | 0.5 |
| 26.01.19 | 1.1 | 0.3 | 0.1 | 3.8 | 1.5 | 1.6 | -11.1 | 0.00 | 0.20 | 0.40 | 0.29 | 0.7 |
| 05.02.19 | 1.5 | 0.2 | 0.0 | 4.3 | 0.1 | 0.7 | -9.6 | 0.00 | 0.30 | 0.62 | 0.46 | 0.8 |
| 03.03.19 | 3.8 | 1.2 | 0.1 | 10.4 | 1.7 | 1.3 | -8.4 | 0.04 | 0.30 | 0.82 | 0.54 | 0.9 |
| 08.03.19 | 2.3 | 0.8 | 0.1 | 6.6 | 2.6 | 3.6 | -13.8 | 0.04 | 0.45 | 0.86 | 0.66 | 0.8 |
| 27.03.19 | 1.2 | 0.2 | 0.0 | 3.9 | -0.2 | 1.3 | -8.7 | 0.00 | 0.22 | 0.48 | 0.34 | 1.3 |
| 07.05.19 | 2.2 | 0.1 | 0.1 | 6.3 | 1.8 | 2.0 | -7.7 | 0.00 | 0.37 | 0.92 | 0.58 | 1.9 |
| 14.05.19 | 2.9 | 0.3 | 0.2 | 7.0 | 1.8 | 5.6 | -7 | 0.00 | 0.29 | 1.01 | 0.51 | 2.3 |
| 03.09.19 | 2.6 | 1.6 | 0.0 | 5.6 | 2.9 | 1.6 | 1.3 | 0.00 | 0.35 | 0.75 | 0.48 | 1.4 |
| 12.09.19 | 1.7 | 1.4 | 0.0 | 3.1 | 3.6 | 4.3 | 1.2 | 0.13 | 0.21 | 0.99 | 0.48 | 1.1 |
| 16.10.19 | 3.3 | 1.1 | 0.0 | 5.3 | 2.2 | 2.1 | -4.9 | 0.13 | 0.43 | 0.69 | 0.56 | 0.6 |
| 28.12.19 | 3.3 | 0.3 | 0.1 | 7.3 | 1.3 | 2.0 | -7 | 0.00 | 0.43 | 0.94 | 0.64 | 0.7 |
| 30.12.19 | 1.6 | 0.4 | 0.0 | 2.9 | 0.1 | 1.1 | -2.4 | 0.00 | 0.38 | 1.01 | 0.68 | 0.4 |
| 05.01.20 | 1.7 | 0.5 | 0.1 | 7.5 | 0.1 | 1.2 | -7.5 | 0.00 | 0.26 | 0.78 | 0.43 | 0.5 |
| 14.01.20 | 4.0 | 0.6 | 0.0 | 10.5 | 1.4 | 1.0 | -6.1 | 0.00 | 0.54 | 0.84 | 0.68 | 1.1 |
| 16.02.20 | 4.2 | 0.6 | 0.2 | 8.9 | 1.3 | 2.0 | -0.9 | 0.00 | 0.23 | 0.66 | 0.49 | 0.8 |
| 20.02.20 | 4.9 | 0.7 | 0.0 | 11.4 | 1.6 | 3.3 | -10.1 | 0.00 | 0.19 | 0.51 | 0.34 | 1.7 |
| 10.03.20 | 1.4 | 0.1 | 0.1 | 7.4 | 0.3 | 1.4 | -8.4 | 0.08 | 0.16 | 0.96 | 0.45 | 2.6 |
| 02.05.20 | 0.7 | 0.3 | 0.0 | 1.9 | 1.6 | 1.0 | -8.8 | 0.00 | 0.48 | 1.04 | 0.79 | 0.5 |
| 03.05.20 | 3.9 | 0.5 | 0.0 | 9.8 | 3.0 | 3.4 | -9.2 | 0.04 | 0.22 | 0.88 | 0.50 | 0.9 |
| 07.07.20 | 2.4 | 2.2 | 0.1 | 3.9 | 6.1 | 6.3 | 0.3 | 0.00 | 0.50 | 1.00 | 0.73 | 1.3 |
| 31.10.20 | 1.8 | 0.3 | 0.2 | 4.9 | 2.5 | 1.8 | -0.9 | 0.00 | 0.48 | 0.98 | 0.66 | 0.7 |
| 01.11.20 | 1.7 | 0.5 | 0.2 | 6.8 | 2.6 | 0.6 | -1.3 | 0.42 | 0.26 | 0.64 | 0.46 | 0.6 |
| 22.11.20 | 0.9 | 0.5 | 0.0 | 4.4 | 0.9 | 0.7 | -2.3 | 0.00 | 0.31 | 0.96 | 0.56 | 0.7 |
| 22.12.20 | 2.2 | -0.1 | 0.0 | 5.5 | 0.1 | 1.8 | -7 | 0.00 | 0.15 | 0.35 | 0.22 | 0.4 |
| 19.01.21 | 4.1 | 0.0 | 0.0 | 12.8 | -0.1 | 1.6 | -13 | 0.00 | 0.34 | 0.64 | 0.49 | 1.6 |
| 14.02.21 | 0.7 | 0.2 | 0.1 | 5.7 | 0.9 | 0.9 | -17 | 0.00 | 0.30 | 0.62 | 0.40 | 0.5 |
| 14.03.21 | 0.5 | 0.5 | 0.1 | 1.5 | 2.3 | 0.4 | -18.8 | 0.83 | 0.57 | 0.90 | 0.75 | 1.7 |
| *Saharan dust intrusions* | | | | | | | | | | | | |
| 25.01.20 | 2.4 | -0.1 | 0.5 | 5.7 | 1.0 | 2.1 | -11.0 | 0.00 | 2.10 | 4.99 | 3.01 | 20.8 |
| 03.02.20 | 1.5 | 0.9 | 0.1 | 4.5 | 1.9 | 5.1 | -5.9 | 0.00 | 1.17 | 2.31 | 1.68 | 8.7 |
| 21.03.20 | 4.8 | 0.3 | 0.1 | 13.9 | 5.1 | 9.4 | -8.8 | 0.00 | 1.24 | 5.03 | 2.71 | 16.5 |
| 22.03.20 | 1.2 | 0.2 | 0.0 | 13.0 | 9.9 | 5.6 | -9.2 | 0.00 | 0.46 | 5.14 | 3.60 | 9.6 |
| 08.11.20 | 2.3 | -0.1 | 0.0 | 12.7 | 1.5 | 6.8 | -4.3 | 0.00 | 1.37 | 4.06 | 2.36 | 13.8 |
| 06.02.21 | 1.5 | 1.2 | 0.4 | -25.1 | -37.8 | 116.0 | -7.2 | 0.00 | 1.15 | 3.91 | 2.47 | 147.1 |
| 03.03.21 | 0.4 | 0.5 | 0.1 | -0.2 | 6.8 | 12.4 | -7.1 | 0.00 | 1.25 | 2.40 | 1.58 | 61.5 |

**Table A2: Number concentration of ice nucleating particles active at - 10 °C (INP$_{-10}$) at Jungfraujoch, determined on randomly selected PM$_{10}$ filter samples. Each sample was collected throughout a full calender day, for which local mean air temperature and the fraction of hours with relative humidity above 95% is shown. The INP data are from the study by Conen et al. (2015) and meteorological data is from MeteoSwiss. Assays of INP that showed no freezing at all at -10 °C were set to a value of 0.01 INP$_{-10}$ m$^{-3}$ to be displayable on the log-scale in Figure 2.**

| date | INP$_{-10}$ | T$_{mean}$ | fraction of hours with RH > 95% | date | INP$_{-10}$ | T$_{mean}$ | fraction of hours with RH > 95% |
|---|---|---|---|---|---|---|---|
| | (m$^{-3}$) | (°C) | | | (m$^{-3}$) | (°C) | |
| 06.06.12 | 2.23 | -1.7 | 0.17 | 04.12.12 | 1.00 | -16.6 | 0.08 |
| 13.06.12 | 0.78 | -8.1 | 0.54 | 07.12.12 | 0.36 | -18.1 | 0.00 |
| 16.06.12 | 6.43 | 2.2 | 0.00 | 08.12.12 | 0.43 | -21.1 | 0.00 |
| 23.06.12 | 3.61 | 1.2 | 0.00 | 14.12.12 | 0.50 | -9.9 | 0.00 |
| 27.06.12 | 8.78 | 1.7 | 0.17 | 24.12.12 | 0.50 | -3.7 | 0.00 |
| 30.06.12 | 6.94 | 4.4 | 0.08 | 28.12.12 | 3.19 | -13.7 | 0.42 |
| 03.07.12 | 19.50 | 1.7 | 0.25 | 01.01.13 | 0.71 | -10.7 | 0.21 |
| 08.07.12 | 8.78 | 0.7 | 0.08 | 11.01.13 | 0.57 | -16.2 | 0.04 |
| 11.07.12 | 3.73 | -1.1 | 0.04 | 19.01.13 | 0.09 | -9.5 | 0.04 |
| 13.07.12 | 4.07 | -0.9 | 0.17 | 20.01.13 | 0.03 | -11.9 | 0.13 |
| 16.07.12 | 2.05 | -5 | 0.25 | 22.01.13 | 0.01 | -18.8 | 0.00 |
| 28.07.12 | 15.38 | 2 | 0.17 | 26.01.13 | 0.01 | -14.2 | 0.00 |
| 01.08.12 | 5.50 | 3.2 | 0.17 | 01.02.13 | 0.92 | -9.5 | 0.54 |
| 03.08.12 | 20.74 | 1.8 | 0.00 | 06.02.13 | 0.85 | -21.3 | 0.96 |
| 10.08.12 | 2.60 | -0.8 | 0.17 | 09.02.13 | 0.03 | -26.8 | 0.54 |
| 15.08.12 | 5.95 | 3 | 0.17 | 11.02.13 | 0.16 | -17.4 | 0.00 |
| 18.08.12 | 3.95 | 7.4 | 0.00 | 21.02.13 | 0.01 | -20 | 0.13 |
| 20.08.12 | 6.10 | 5.8 | 0.13 | 26.02.13 | 0.01 | -18 | 0.00 |
| 03.09.12 | 4.55 | -0.9 | 0.21 | 04.03.13 | 0.93 | -10.6 | 0.00 |
| 15.09.12 | 2.14 | 0.6 | 0.00 | 06.03.13 | 0.10 | -10.8 | 0.25 |
| 19.09.12 | 1.80 | -7 | 0.75 | 13.03.13 | 0.23 | -15.5 | 0.00 |
| 21.09.12 | 1.39 | -0.9 | 0.04 | 14.03.13 | 0.36 | -25.4 | 1.00 |
| 25.09.12 | 12.03 | -3.9 | 0.13 | 18.03.13 | 0.10 | -16.3 | 0.00 |
| 30.09.12 | 4.68 | -2.9 | 0.79 | 31.03.13 | 0.30 | -17.9 | 0.00 |
| 04.10.12 | 2.32 | -4.4 | 0.29 | 10.04.13 | 0.78 | -11.8 | 0.00 |
| 11.10.12 | 2.05 | -4.3 | 0.04 | 11.04.13 | 0.50 | -6.3 | 0.00 |
| 12.10.12 | 3.29 | -6.1 | 0.29 | 14.04.13 | 1.16 | -2.3 | 0.00 |
| 16.10.12 | 0.64 | -8 | 0.00 | 19.04.13 | 2.79 | -7.6 | 0.46 |
| 23.10.12 | 2.14 | 1.8 | 0.00 | 23.04.13 | 0.36 | -4.5 | 0.00 |
| 28.10.12 | 0.03 | -16.5 | 0.00 | 29.04.13 | 0.43 | -4.1 | 0.92 |
| 02.11.12 | 3.51 | -10.6 | 0.00 | 05.05.13 | 1.88 | -5.9 | 0.00 |
| 10.11.12 | 9.02 | -7.1 | 0.58 | 17.05.13 | 0.86 | -5.2 | 0.50 |
| 13.11.12 | 1.88 | -4.3 | 0.00 | 22.05.13 | 0.03 | -9.1 | 0.00 |
| 14.11.12 | 11.31 | -2 | 0.00 | 25.05.13 | 0.50 | -15.5 | 0.67 |
| 21.11.12 | 0.23 | -5.8 | 0.00 | 28.05.13 | 1.80 | -7 | 0.63 |
| 26.11.12 | 0.86 | -6.9 | 0.63 | 29.05.13 | 0.30 | -12.4 | 0.33 |



**Table A3: Median number concentration of ice nucleating particles active between -8 °C and -15 °C at Jungfraujoch, determined for free tropospheric conditions and during Saharan dust intrusions, and average of two assays with blank filter material. The blank values have been subtracted from the sample values before calculating the medians shown in the Table.**


| Temperature (°C) | -8 | -9 | -10 | -11 | -12 | -13 | -14 | -15 |
|---|---|---|---|---|---|---|---|---|
| Category | INP in free troposphere (m$^{-3}$) | | | | | | | |
| all | 0.8 | 1.7 | 2.4 | 3.4 | 4.4 | 6.0 | 8.7 | 9.8 |
| heat sensitive | 0.6 | 1.4 | 2.0 | 2.5 | 3.4 | 4.1 | 4.9 | 5.6 |
| moderately heat tolerant | 0.3 | 0.3 | 0.4 | 0.5 | 0.6 | 0.9 | 1.2 | 1.5 |
| heat tolerant | 0.0 | 0.0 | 0.0 | 0.2 | 0.3 | 0.5 | 0.9 | 1.6 |
| | INP in Saharan dust intrusions (m$^{-3}$) | | | | | | | |
| all | 0.6 | 1.3 | 2.4 | 3.2 | 5.1 | 7.8 | 13.4 | 20.0 |
| heat sensitive | 0.5 | 1.0 | 1.9 | 2.7 | 4.5 | 5.1 | 6.9 | 9.2 |
| moderately heat tolerant | 0.3 | 0.2 | 0.2 | 0.2 | 0.7 | 1.0 | 1.9 | 3.5 |
| heat tolerant | 0.0 | 0.0 | 0.1 | 0.3 | 0.5 | 1.1 | 3.3 | 6.2 |
| | INP in blank filter material (m$^{-3}$) | | | | | | | |
| all | 0.0 | 0.0 | 0.1 | 0.1 | 0.2 | 0.3 | 0.4 | 0.6 |
| heat sensitive | 0.0 | 0.0 | 0.0 | 0.0 | 0.0 | 0.2 | 0.2 | 0.2 |
| moderately heat tolerant | 0.0 | 0.0 | 0.1 | 0.1 | 0.1 | 0.0 | 0.0 | 0.1 |
| heat tolerant | 0.0 | 0.0 | 0.0 | 0.0 | 0.0 | 0.1 | 0.1 | 0.4 |