# Peer review of "Measurement report: Ice nucleating particles active $\geq -15$ °C in free tropospheric air over western Europe"

_Atmospheric Chemistry and Physics, 2021_

## Referee Comment (RC1)

In this measurement report, the authors present ice nucleation data of aerosols sampled on filters between January 2019 and March 2021 at the high-altitude research station Jungfraujoch in Switzerland. The authors present concentrations of ice nucleating particles active above a supercooling temperature of -10°C and -15°C in the lower free troposphere (FT). Radon isotope data was used to distinguish between the FT and planetary boundary layer (PBL). The authors chose to perform a heat treatment experiment to study the composition of the aerosol samples and they discuss the influence of occasional PBL events (Sahara dust) on the INP concentration. The work presented merits publication in Atmospheric Chemistry and Physics after minor revisions.

In general, the study provides further evidence that the role of biological INPs to ice formation in the lower troposphere should not be underestimated. However, one major concern to me is that the authors only use heat treatment tests to evaluate the composition of aerosols. Heat treatment experiments have gained acceptance in the scientific community to broadly categorize INPs. However, heat tests alone do not identify certain microorganism. Have you considered additional investigations of the studied samples to get more thorough information (e.g. cultivating microorganisms)? If not, alternative methods should be mentioned in the paper and the limitation of the results needs to be discussed in more detail.

**Introduction:**

The introduction is written very clearly. However, the section would be improved if the authors would elaborate more on aerosol transportation. The authors suggest epiphytic microorganisms (MO) to be the major INP. How do you think MO are transported from the PBL to the FT? In last years, many studies have suggested that rain events over vegetation lead to an increase in bio-INP concentration (e.g. Huffman et al.2013, Prenni et al., 2013, Iwata et al., 2019). Do you think these events are important for the transportation of MO to the free troposphere?

In line 36 the authors write: *"[…] at more moderate supercooling (here ≥ -15°C) biological particles constitute the main part of the INP population, at least in the PBL."* Doesn't that strongly depend on the location? There are also mineral particles which nucleate ice at temperatures >-15°C (see e.g. Harrison et al., 2016). Please elaborate more clearly.

**Materials and methods:**

*2.1.*

In total 72 punches were analysed for their ice nucleation activity. For me it is unclear, how many punches per filter were used for the ice nucleation measurements? Could you please clarify this in the manuscript?

Additionally, I am wondering whether such large filter pieces influence the measurement. The paper would greatly benefit if results of a background measurement are provided in the appendix.

The authors state that the ice nucleation data was adjusted with a background correction. Was this done in accordance to Vali, 2019?

Could you please clarify what filter-fringes are? (line 59)

**Results and discussion**

My main criticism on the paper is the identification of microorganisms, based only on heat treatment experiments. How can you exclude that repeated freeze-thaw cycles be partially responsible for the loss of ice nucleation activity (see e.g. Polen et al., 2016)? Please discuss more clearly.

Table 1: The table would be improved if the authors could write the amount of analysed samples to each category. In addition, I was wondering why the sum of the categories *heat sensitive, moderately heat tolerant* and *heat tolerant* do not reach the value of the category *all*? The table caption starts with an explanation of the standard deviations; wouldn't it be better to first mention the INP concentration?

Table 2 gives a very good overview of the reported heat sensitivity of INPs in the literature.

Line 145: Do you think that impingers would be more efficient to sample and analyse INPs in general? Since INPs only occur in low concentrations in the FT, filter pose a challenge that an impinger may be able to handle. With an impinger, INPs are directly impacted in a solution which can be used in a freezing assay and the intermediate step of suspending INPs from a filter is avoided. Could you maybe discuss this issue in more detail?

Figure 3. In contrast to the previous study, the authors do not see a seasonal trend in their data. However, in this study only one data point was considered during summer time. This seems too weak to clearly rule out seasonal dependency. I would encourage the authors to lower the strength of the argument.

*Technical comments:*

Line 27: A minus sign is missing [...] "-"5°C [...]

Line 30: Another minus sign is missing [...] "-"31°C [...]

Line 30: This sentence requires a citation. Which other regions? Please specify.

Line 35: Another minus sign is missing [...] "-"15°C [...]

Line 46: Superscript number 3: "[…] air was sampled at a rate of 720 m"$^3$" day$^{-1}$

Line 60: It should be "Sample values […]" not Sample"s"

Line 81: Saharan dust intrusion (SD) – the abbreviation was already introduced in the introduction (line 38).

Line 102: "*Fusarium graminearum*" and "*Puccinia sp.*" should be written in italic

Line 105: Maybe reconsider the sentence position. We found …

Line 106: "*Mortierella alpine*" should be written in italic

Line 123: change phrase to"[…] but excluded *one sample* from further analysis […]"

Line 373: Superscript -3: "[…] 0.095 INP m$^{-3}$"

Line 440: Remove space between the minus sign and the number "-10 °C"

Line 444: It should be Figure "3"

**References:**

Harrison, Alexander D., et al. "Not all feldspars are equal: a survey of ice nucleating properties across the feldspar group of minerals." *Atmospheric Chemistry and Physics* 16.17 (2016): 10927-10940.

Huffman, J. Alex, et al. "High concentrations of biological aerosol particles and ice nuclei during and after rain." Atmospheric Chemistry and Physics 13.13 (2013): 6151-6164.

Prenni, A.J.; Tobo, Y.; Garcia, E.; DeMott, P.J.; Huffman, J.A.; McCluskey, C.S.; Kreidenweis, S.M.; Prenni, J.; Pöhlker, C.; Pöschl, U.The impact of rain on ice nuclei populations at a forested site in Colorado. Geophys. Res. Lett. 2013, 40, 227–231

Iwata, A.; Imura, M.; Hama, M.; Maki, T.; Tsuchiya, N.; Kunihisa, R.; Matsuki, A. Release of Highly Active Ice Nucleating Biological Particles Associated with Rain. Atmosphere 2019, 10, 605.

Vali, G. Revisiting the differential freezing nucleus spectra derived from drop-freezing experiments: Methods of calculation,applications, and confidence limits. Atmos. Meas. Tech. 2019, 12, 1219–1231.

Polen, M., Lawlis, E., & Sullivan, R. C. The unstable ice nucleation properties of Snomax® bacterial particles. Journal of Geophysical Research: Atmospheres, 2016, 121(19), 11-666.

---

## Referee Comment (RC2)

Review

Measurement report: Ice nucleating particles active ≥ -15 °C in free tropospheric air over western Europe by Conen et al.

Summary

The study presented by F. Conen et al. reports INP concentrations at -10 °C and -15 °C at the High Altitude Research Station Jungfraujoch, a site which is located in the lower free troposphere. The measurements are based on 24-hour collected filters throughout several years, and focus on sampling intervals when the site was not impacted by boundary layer air. By performing heat treatment tests, it is suggested that the majority of INPs in this temperature regime are microorganisms at Jungfraujoch. While this is an important finding and good motivation for future studies, I have some concerns about the methods used and the conclusions drawn from them.

General remarks

- One of the key points in your study is the distinction between free tropospheric conditions and boundary layer intrusion. However, you use only one method to quantify this, namely the radon concentration. In the study by Herrmann et al. (2015) it was found that "local radon emissions as well as the comparatively long radon lifetime blur the distinction between free tropospheric conditions and boundary layer influence.This further supports the conclusion that the CO/NOy approach offers the best distinction between free troposphere conditions and boundary layer influence." Is there a reason why you did not consider the ratio between CO and NOy to identify free tropospheric conditions, e.g., in addition to the Radon concentration?
- The identification of Saharan dust impact is solely based on "high PM10 loads" (lines 80 – 81). Is there a study showing that such measures can be used to determine Saharan dust events at Jungfraujoch? If not, you need to provide more results on this approach, as for example a comparison to other methods used to dermine Saharn dust impact, such as aerosol particle scattering properties (Collaud Coen et al., 2004), occurrence of larger particles (e.g., Kammermann et al., 2010) and back trajectory analysis.
- As discussed by the authors, the 24-hr filter collection are limited in their use to distinguish free tropospheric from disturbed free tropospheric conditions. Did the authors consider performing shorter filter sampling times?
- The study comprises only 28 datapoints of INP concentrations, which is not much to draw general conclusions about seasonal variability.
- You do not present the freezing spectra but only INP concentrations at specific temperatures. It might be interesting to the reader to see those spectra, e.g., in the appendix.

Abstract

Lines 7 – 8: The statement "… typically starting a few km above ground" is not correct. Ice formation in mixed-phase clouds can occur at ground level; e.g., Jungfraujoch is per definition a ground-based station and primary ice formation frequently occurs there (e.g., Mertes et al., 2007), which is also true for other locations in mountains. Thus, ice formation in clouds is rather a question of temperature, supersaturation, and presence of INPs, not of altitude above ground.

Lines 16 - 18: I am struggling with this last statement. Jungfraujoch is a site which can be impacted by boundary layer intrusions, which can occur on short time scales. How certain are you, that you excluded such potential boundary layer intrusions from your 24 hr filters (see also my comment above)? In addition, Jungfraujoch is impacted by touristic activities (e.g., smoking), and your 24-hour filters are impacted by this. Has one filter been sampled without any impact from touristic activities, e.g., during a lockdown in 2020 due to the COVID pandemic? Generally, to make such a strong statement, more of such measurements should be conducted with shorter filter collection times (to

better avoid boundary layer intrusions or emissions from touristic activities), and at other free-tropospheric sites over central/Western Europe to confirm this. Your study is a good motivation for such future research.

I recommend to specify the time period during which you collected your samples (months, years).

Introduction

Lines 22 – 24: Please check the grammatic of this sentence. I am unsure if I understand the sentence correctly. Do you mean that this community of airborne microorganisms have their emission sources within several tens of kilometers (local/regional scale)?

Line 27: The statement regarding altitude for mixed-phase cloud occurrence is not correct, see my comment before.

Line 28: Please add more citations to this very important statement. E.g., many airborne INP measurements are conducted in the free troposphere, and there are more free-tropospheric measurements sites where INP studies were conducted.

Lines 32 - 34: The cited percentages refer to ice particle residuals, not to the INP concentration measurements at -31°C.

Lines 35 – 36: I recommend to also include citations from ice particle residual analysis performed at Jungfraujoch (e.g., Mertes et al., 2007; Kupiszewski et al., 2015; Schmidt et al., 2017; Lacher et al., 2021).

Material and methods

I recommend to specify the time period during which you collected your samples (months, years) and to refer to the respective table in the appendix.

Line 52: What is the error associated to a counting by eye?

Lines 71: I recommend to also cite Herrmann et al. (2015) here.

Lines 80 – 81: Please provide more information about the method used to identify impact from Saharan dust event. Which threshold has been used? Has such a method been established in previous work, resp., has it been validated against other measurements or back trajectory calculations (see also my comment above)?

 Line 82: It is not clear to me why you analyzed the median radon concentration here, while before (line 76) the mean radon concentration was used to identify free tropospheric conditions.

Results and discussion

Lines 128 – 135: Your discussion is based on the assumption that during Saharan dust events all aerosol particles are dust particles. With the methods used here you cannot quantify the fraction of dust particles in the overall particle population.

Lines 151 – 153: During Saharan dust events, the total number concentration of aerosol particles does not necessarily increase; more important here would be to quantify the concentration of dust particles and compare it to an increase in INP concentration.

Technical

Line 27: I assume you mean "-5 °C".

Line 30: It should read "-31 °C".

Line 45: I assume it should read "At the site,…"?

Line 60: "Sample" instead of "Samples"?

References

Collaud Coen, M., Weingartner, E., Schaub, D., Hueglin, C., Corrigan, C., Henning, S., Schwikowski, M., and Baltensperger, U.: Saharan dust events at the Jungfraujoch: detection by wavelength dependence of the single scattering albedo and first climatology analysis, Atmos. Chem. Phys., 4, 2465-2480, 10.5194/acp-4-2465-2004, 2004.

Kammermann, L., Gysel, M., Weingartner, E., and Baltensperger, U.: 13 month climatology of the aerosol hygroscopicity at the free tropospheric site Jungfraujoch (3580 m a.s.l.), Atmos. Chem. 95 Phys., 10, 10717–10732, https://doi.org/10.5194/acp-10-10717-2010, 2010.

Kupiszewski, P., Zanatta, M., Mertes, S., Vochezer, P., Lloyd, G., Schneider, J., Schenk, L., Schnaiter, M., Baltensperger, U., Weingartner, E., and Gysel, M.: Ice residual prop   erties in mixed-phase clouds at the high-alpine Jungfrau   joch site, J. Geophys. Res.-Atmos., 121, 12343–12362, https://doi.org/10.1002/2016JD024894, 2016.

Lacher, L., Clemen, H.-C., Shen, X., Mertes, S., Gysel-Beer, M., Moallemi, A., Steinbacher, M., Henne, S., Saathoff, H., Möhler, O., Höhler, K., Schiebel, T., Weber, D., Schrod, J., Schneider, J., and Kanji, Z. A.: Sources and nature of ice-nucleating particles in the free troposphere at Jungfraujoch in winter 2017, Atmos. Chem. Phys., 21, 16925–16953, https://doi.org/10.5194/acp-21-16925-2021, 2021.

Mertes, S., Verheggen, B., Walter, S., Connolly, P., Ebert, M., Schneider, J., Bower, K. N., Cozic, J., Weinbruch, S., Baltensperger, U., and Weingartner, E.: Counterflow Virtual Impactor Based Collection of Small Ice Particles in Mixed-Phase Clouds for the Physico-Chemical Characterization of Tropospheric Ice Nuclei: Sampler Description and First Case Study, Aerosol Sci. Tech., 41, 848–864, https://doi.org/10.1080/02786820701501881, 2007.

Schmidt, S., Schneider, J., Klimach, T., Mertes, S., Schenk, L. P., Kupiszewski, P., Curtius, J., and Borrmann, S.: Online single particle analysis of ice particle residuals from mountain-top mixedphase clouds using laboratory derived particle type assignment, Atmos. Chem. Phys., 17, 575–594, https://doi.org/10.5194/acp-17-575-2017, 2017.

---

## Author Comment (AC1)

Response to RC1

(comments in black, response in blue, *changes made to manuscript in italic [page and line numbers refer to revised version]*)

In this measurement report, the authors present ice nucleation data of aerosols sampled on filters between January 2019 and March 2021 at the high-altitude research station Jungfraujoch in Switzerland. The authors present concentrations of ice nucleating particles active above a supercooling temperature of -10°C and -15°C in the lower free troposphere (FT). Radon isotope data was used to distinguish between the FT and planetary boundary layer (PBL). The authors chose to perform a heat treatment experiment to study the composition of the aerosol samples and they discuss the influence of occasional PBL events (Sahara dust) on the INP concentration. The work presented merits publication in Atmospheric Chemistry and Physics after minor revisions.

In general, the study provides further evidence that the role of biological INPs to ice formation in the lower troposphere should not be underestimated. However, one major concern to me is that the authors only use heat treatment tests to evaluate the composition of aerosols. Heat treatment experiments have gained acceptance in the scientific community to broadly categorize INPs. However, heat tests alone do not identify certain microorganism. Have you considered additional investigations of the studied samples to get more thorough information (e.g. cultivating microorganisms)? If not, alternative methods should be mentioned in the paper and the limitation of the results needs to be discussed in more detail.

We are grateful to the reviewer for having read our manuscript and for the overall judgement.

Indeed, the use of heat tests provides only for a coarse categorisation of INP, whereas cultivation methods are more precise but also require much larger sample volumes. In an earlier study at Jungfraujoch, Stopelli et al. (2017) have targeted by selective cultivation *Pseudomonas syringae* in freshly fallen snow. In only 3 of 13 samples P. syringae was found in form of 2, 4 and 45 colony-forming units per litre of melted snow. The maximum of these three values corresponds to 0.02 colony forming units in 1 m$^{-3}$ of air (45 colony forming units / 1000 g water * 0.4 g water m$^{-3}$ air), assuming a condensed water content of 0.4 g m$^{-3}$ (Petters and Wright, 2015). Overall, cultivable cells of *P. syringae* constituted a tiny fraction (10$^{-4}$) of all INP$_{-8}$ in the investigated snow samples.

Apart from the limitations imposed by sample volume, INP in the atmosphere probably include a substantial fraction of cell-free INP that are not cultivable, such as macromolecules shed by hyphae of *Mortierella alpina* (Fröhlich-Nowoisky et al., 2015).

Introduction:

The introduction is written very clearly. However, the section would be improved if the authors would elaborate more on aerosol transportation. The authors suggest epiphytic microorganisms (MO) to be the major INP. How do you think MO are transported from the PBL to the FT? In last years, many studies have suggested that rain events over vegetation lead to an increase in bio-INP concentration (e.g. Huffman et al.2013, Prenni et al., 2013, Iwata et al., 2019). Do you think these events are important for the transportation of MO to the free troposphere?

Yes, we definitively think the aerosolisation of MO is intensified by rain events. When rain comes with a frontal system or a thunderstorm part of the MO are probably mixed with or injected to the

FT. Another possibility is injection to the FT from PBL air masses advected to mountains. We have added the following text near the end of the introduction:

*Aerosolisation of biological particles from vegetated land surface to the PBL is intensified during rainfall (Huffman et al. 2013; Prenni et al., 2013; Iwata et al., 2019). Subsequent transport from the PBL to the FT may happen through mixing in frontal systems, cloud convection, or by mountain venting (Henne et al., 2005).   [page 2, lines 51 to 53]*

In line 36 the authors write: "[…] at more moderate supercooling (here ≥ -15°C) biological particles constitute the main part of the INP population, at least in the PBL." Doesn't that strongly depend on the location? There are also mineral particles which nucleate ice at temperatures >-15°C (see e.g. Harrison et al., 2016). Please elaborate more clearly.

We do not exclude that also mineral particles contribute to the INP population in the PBL. Yet, heat tests suggest they constitute a minor fraction even in dust events. The debatable part of the introduction is changed accordingly, also in response to a comment made by Reviewer #2:

*Heat deactivates biological INP but leaves mineral INP largely unaffected (Hill et al., 2016). Chen et al. (2021) found during dust events sampled in Beijing that 70% of INP active at ≥ -15 °C (INP$_{-15}$) were heat-sensitive. Observed heat-sensitive fractions of INP$_{-15}$ in the PBL of agricultural areas in the USA (Suski et al., 2018) and in Argentina (Testa et al., 2021) were even larger (> 90%). These findings contrast with a small biological fraction detected among ice particle residuals collected at Jungfraujoch from mixed-phase clouds and classified by physico-chemical analyses (Mertes et al., 2007) or by laser ablation mass-spectrometry (Schmidt et al., 2017; Lacher et al., 2021). Not every ice particle residual has initiated as INP the formation of the ice particle it was recovered from, in particular not when it was recovered from a secondary ice particle. An ice particle residual recovered from a primary ice particle and classified by mass spectrometry as mineral dust may have been activated at moderate supercooling by a minor, ice-nucleation active biological component sticking to its surface (Augustin-Bauditz et al., 2016). A heat test, which specifically targets the ice-nucleation active component, would have classified the same assembly as biological INP.   [page 1, line 38 to page 2, line 49]*

Materials and methods:

2.1.

In total 72 punches were analysed for their ice nucleation activity. For me it is unclear, how many punches per filter were used for the ice nucleation measurements? Could you please clarify this in the manuscript?

The issue was clarified by changing the sentence:

*From each selected filter we took 72 punches with 2 mm diameter each; all 72 punches together contained aerosol particles from a total of 10.6 m$^{-3}$.   [page 2, lines 66 and 67]*

Additionally, I am wondering whether such large filter pieces influence the measurement. The paper would greatly benefit if results of a background measurement are provided in the appendix.

We have added results of the background (blank) measurements to the appendix (Table A3). They show that the filter material has little influence on the measurement. To the first paragraph of the Results and discussion section we have added:

*A majority of INP$_{-10}$ (83%) and IN$_{P-15}$ (57%) in the FT were heat sensitive and lost their activity after exposure to 60 °C (Table 1; for INP active at other temperatures between -8 °C and -15 °C see Table A3).   [page 4, lines 125 and 126; added Table on page 18]*

The authors state that the ice nucleation data was adjusted with a background correction. Was this done in accordance to Vali, 2019?

The background correction according to Vali (2019) is a powerful tool when analysing freezing spectra. In our manuscript we focus on INP active at two temperatures only, -10 °C and -15 °C. Therefore, we have simply subtracted the cumulative blank value from the cumulative sample value at either temperature.

Could you please clarify what filter-fringes are? (line 59)

We have expanded the description accordingly:

*Two blank (background) assays were done with punches from 5 mm wide fringes of sample filters. This part of a filter is covered by the clamp rings holding it in place during sampling. That way, sampled air does not pass through it and it remains clean. However, during handling and transport some particles may get smeared from the active filter area onto this narrow fringe. Consequently, these blank values are a conservative (upper) estimate of a field blank. Each of these blanks was composed of punches from four filters. Sample values were corrected for blank values by subtracting the average of both blanks, which on average was 7% of a sample value.   [page 3, lines 82 to 89]*

Results and discussion

My main criticism on the paper is the identification of microorganisms, based only on heat treatment experiments. How can you exclude that repeated freeze-thaw cycles be partially responsible for the loss of ice nucleation activity (see e.g. Polen et al., 2016)? Please discuss more clearly.

This is an important issue. We think it is best discussed in the materials and methods section. We have added several lines to the end of its first paragraph:

*Loss of INP active < -10 °C due to the repeated freezing is unlikely in this study. Although Polen et al. (2016) have observed throughout five repeated freezing assays some loss of INP active > -5 °C, little had changed in INP active < -5 °C (see Fig. 6 in Polen et al., 2016). A more extensive set of tests was conducted by Vali (2008) on a soil sample. After 55 refreezing cycles an increasingly larger fraction of INP had been lost above -10 °C toward the warmer end of the freezing spectrum, but the concentration of INP active < -10 °C had remained practically the same (Fig. 1c in Vali, 2008). [page 2, lines 72 to 77]*

Table 1: The table would be improved if the authors could write the amount of analysed samples to each category. In addition, I was wondering why the sum of the categories heat sensitive, moderately heat tolerant and heat tolerant do not reach the value of the category all? The table caption starts with an explanation of the standard deviations; wouldn't it be better to first mention the INP concentration?

We have added the number of samples analysed in each category. The sum of the three categories does not to match the value of category 'all' because these values are not averages but medians of differently skewed distributions.

We changed the start of the caption to:

*INP concentration found in the free troposphere (FT) and during Saharan dust intrusions (SD), categorised according to heat sensitivity. Median and multiplicative standard deviation are shown. [page 11, lines 429 and 439]*

Table 2 gives a very good overview of the reported heat sensitivity of INPs in the literature.

Thank you for this comment.

Line 145: Do you think that impingers would be more efficient to sample and analyse INPs in general? Since INPs only occur in low concentrations in the FT, filter pose a challenge that an impinger may be able to handle. With an impinger, INPs are directly impacted in a solution which can be used in a freezing assay and the intermediate step of suspending INPs from a filter is avoided. Could you maybe discuss this issue in more detail?

That is correct. A high volume impinger, such as the one used by Mignani et al. (2021) with a flow rate of 300 L/min has the advantages you mention. In addition, it allows for a higher resolution in INP time series. But impingers also have an important disadvantage: They still require the continuous presence of an operator. Maybe, one day someone will develop an automated impinger similar to the automated continuous flow diffusion chamber developed by Brunner and Kanji (2021). Until this has happened, the use of impingers is limited to field campaigns. In contrast, the filters we have been using are the result of a continuous, ongoing monitoring effort and are available for almost every day of many years back. The sampling unit requires a little attention only once a fortnight. With the filter archive we can address questions arising after the events we eventually become interested in have happened. Impingers and filters are complementary approaches. We think the filters have well served the purpose of investigating INP composition throughout a longer time period in FT and during several SD events.

Figure 3. In contrast to the previous study, the authors do not see a seasonal trend in their data. However, in this study only one data point was considered during summer time. This seems too weak to clearly rule out seasonal dependency. I would encourage the authors to lower the strength of the argument.

We have changed the contentious beginning of the paragraph:

*A surprising feature of INP$_{-10}$ in FT samples is their narrow distribution (1.0 to 5.6 m$^{-3}$) throughout the year (Fig. 3), which is in contrast to what we had found earlier at Jungfraujoch in filters not selected for FT conditions that covered a range of three orders of magnitude (Conen et al., 2015). [page 5, lines 185 to 187]*

Technical comments:

Line 27: A minus sign is missing [...] "-"5°C [...]

Corrected.

Line 30: Another minus sign is missing [...] "-"31°C [...]

Corrected.

Line 30: This sentence requires a citation. Which other regions? Please specify.

Completed the sentence:

*In FT conditions at the high-altitude observatory Jungfraujoch (3580 m a.s.l.) in the Swiss Alps, Lacher et al. (2018) found similar concentrations of INP active at -31 °C ($INP_{-31}$) as had been reported for the FT in other regions of the world (summarised in Table 2 and Figure 6 in Lacher et al., 2018), and little seasonal variation.   [page 1, lines 30 to 33]*

Line 35: Another minus sign is missing […] "-"15°C […]

Corrected.

Line 46: Superscript number 3: "[…] air was sampled at a rate of 720 m"³" day-1

Corrected.

Line 60: It should be "Sample values […]" not Sample"s"

Corrected.

Line 81: Saharan dust intrusion (SD) – the abbreviation was already introduced in the introduction (line 38).

Corrected.

Line 102: "Fusarium graminearum" and "Puccinia sp." should be written in italic

Done.

Line 105: Maybe reconsider the sentence position. We found …

Done.

Line 106: "Mortierella alpine" should be written in italic

Done.

Line 123: change phrase to"[…] but excluded one sample from further analysis […]"

Done.

Line 373: Superscript -3: "[…] 0.095 INP m-3"

Done.

Line 440: Remove space between the minus sign and the number "-10 °C"

Done.

Line 444: It should be Figure "3"

Corrected.

[revised manuscript text omitted]

---

## Author Comment (AC2)

**Response to RC2**

(comments in black, response in blue, *changes made to manuscript in italic [page and line numbers refer to revised version]*)

**Review**

Measurement report: Ice nucleating particles active  $\geq$  -15 °C in free tropospheric air over western Europe by Conen et al.

**Summary**

The study presented by F. Conen et al. reports INP concentrations at -10 °C and -15 °C at the High Altitude Research Station Jungfraujoch, a site which is located in the lower free troposphere. The measurements are based on 24-hour collected filters throughout several years, and focus on sampling intervals when the site was not impacted by boundary layer air. By performing heat treatment tests, it is suggested that the majority of INPs in this temperature regime are microorganisms at Jungfraujoch. While this is an important finding and good motivation for future studies, I have some concerns about the methods used and the conclusions drawn from them.

Thank you for having read our manuscript, for your general remarks and for the more specific comments.

**General remarks**

- One of the key points in your study is the distinction between free tropospheric conditions and boundary layer intrusion. However, you use only one method to quantify this, namely the radon concentration. In the study by Herrmann et al. (2015) it was found that "local radon emissions as well as the comparatively long radon lifetime blur the distinction between free tropospheric conditions and boundary layer influence. This further supports the conclusion that the CO/NOy approach offers the best distinction between free troposphere conditions and boundary layer influence." Is there a reason why you did not consider the ratio between CO and NOy to identify free tropospheric conditions, e.g., in addition to the Radon concentration?

Right, there are other criteria than radon to draw a line between free tropospheric (FT) conditions and boundary layer influence. Herrmann et al. (2015) had used radon, CO/NOy, and Lagrangian backward simulations to identify the FT particle size distribution at Jungfraujoch and found: *"The fact that the three median FT size distributions are almost identical indicates that the three criteria chose different subsets of data from the available, quite similar FT-like size distributions."* Therefore, we do not think that using another tracer for FT conditions would have resulted in selecting PM10 filters with a different aerosol or INP population. Yet, a different tracer might well have led to a somewhat different subset of filters to choose from.

- The identification of Saharan dust impact is solely based on "high PM10 loads" (lines 80 – 81). Is there a study showing that such measures can be used to determine Saharan dust events at Jungfraujoch? If not, you need to provide more results on this approach, as for example a comparison to other methods used to determine Saharan dust impact, such as aerosol particle scattering properties (Collaud Coen et al., 2004), occurrence of larger particles (e.g., Kammermann et al., 2010) and back trajectory analysis.

In addition to high PM10 loads the SD filters in our study had vivid ochre or brown-yellow colours, similar to the colours of aerial photographs of the Sahara. Filters sampled at Jungfraujoch without SD influence are either white or only lightly coloured in grey or greyish-brown. The ochre or brown-yellow colour is a strong indicator for a substantial SD influence. Collaud Coen et al. (2004) have taken the brown-yellow colouring of filters to confirm the validity of the single scattering exponent as indicator of SD influence. The advantage of the single scattering exponent is that it can detect also weak SD events lasting only few hours, which is too short to produce the filter colouring. Thus, a high PM10 load and a brown-yellow filter colour together reliable indicate a major SD influence. The coincidence of backward trajectories reaching the Sahara in such cases has been shown many times before (e.g. Collaud Coen et al., 2004; Conen et al., 2015). We changed the last sentence of the first paragraph of section 2.2 accordingly:

In addition, we selected from the filter archive an additional seven samples with high PM10 loads and a yellow-brown (ochre) colour resulting from SD intrusions (Collaud Coen et al., 2004). [page 3, lines 107 and 108]

- As discussed by the authors, the 24-hr filter collection are limited in their use to distinguish free tropospheric from disturbed free tropospheric conditions. Did the authors consider performing shorter filter sampling times?

Yes, we have thought about shorter filter sampling times. Apart from filters, also highvolume impingers allow for short sampling times and a reliable detection of INP active > -15 °C (e.g. Mignani et al., 2021), but require the attention of an operator. Hence, their use is currently limited to intensive field campaigns. A shorter filter sampling time with the same setup as used in this study would also demand more labour because filter cassettes would have to be exchanged more often. More importantly, shorter filter sampling times would yield filters with smaller numbers of INP on their surface and a poorer detection limit. Every approach has advantages and trade-offs. The 24 h filter samples were well suited to characterise the INP composition in FT air masses because these filters contain sufficient INP for categorisation by heat tests and they are available for many years back in time. This large pool of available filters compensates for a low fraction (1 in 13) of filters matching the FT criterion.

- The study comprises only 28 datapoints of INP concentrations, which is not much to draw general conclusions about seasonal variability.

We agree. Also in response to a similar comment by Reviewer #1 we have changed the contentious beginning of the paragraph:

A surprising feature of INP-10 in FT samples is their narrow distribution (1.0 to 5.6 m-3) throughout the year (Fig. 3), which is in contrast to what we had found earlier at Jungfraujoch in filters not selected for FT conditions that covered a range of three orders of magnitude (Conen et al., 2015). [page 5, lines 185 to 188]

- You do not present the freezing spectra but only INP concentrations at specific temperatures. It might be interesting to the reader to see those spectra, e.g., in the appendix.

We added median values for all types of spectra (FT, SD, heat sensitivity) in the Appendix (Table A3).

**[additional Table on page 18]**

**Abstract**

Lines 7 – 8: The statement "... typically starting a few km above ground" is not correct. Ice formation in mixed-phase clouds can occur at ground level; e.g., Jungfraujoch is per definition a ground-based station and primary ice formation frequently occurs there (e.g., Mertes et al., 2007), which is also true for other locations in mountains. Thus, ice formation in clouds is rather a question of temperature, supersaturation, and presence of INPs, not of altitude above ground.

Sorry for the imprecision of this statement. It should have been more explicit: "...typically starting in western Europe at a few km above ground." (sentence changed accordingly in revised version; page 1, lines 7 and 8). Although Jungfraujoch is a ground station and often shrouded in mixed-phase clouds, it is not a "typical" location in western Europe. Most of western Europe is located only a few hundred metres above sea level and rarely experiences temperatures below 0 °C. For such a location (Leipzig, Germany) Kanitz et al. (2011, Figure 2 therein) have provided direct evidence in support of this statement (see reply to comment regarding sentence in Introduction Line 27 below).

Lines 16 - 18: I am struggling with this last statement. Jungfraujoch is a site which can be impacted by boundary layer intrusions, which can occur on short time scales. How certain are you, that you excluded such potential boundary layer intrusions from your 24 hr filters (see also my comment above)? In addition, Jungfraujoch is impacted by touristic activities (e.g., smoking), and your 24-hour filters are impacted by this. Has one filter been sampled without any impact from touristic activities, e.g., during a lockdown in 2020 due to the COVID pandemic? Generally, to make such a strong statement, more of such measurements should be conducted with shorter filter collection times (to better avoid boundary layer intrusions or emissions from touristic activities), and at other free-tropospheric sites over central/Western Europe to confirm this. Your study is a good motivation for such future research.

We can never be certain to have completely excluded boundary layer intrusions in a 24 h sampling period. Still, we can have some confidence that it is the case when radon-derived FT "...conditions applied to every hour of a full 24-hour PM10 sampling period." (amended statement in revised version; page 3, line 102). Even if boundary layer influence may have happened on a time scale < 1 h, its overall influence on the sampled INP population must have been very minor because the difference we found in INP-15 concentration between FT and boundary layer influenced SD was only a factor of two. Regarding local aerosol sources, such as cigarette smoke we can say that soot is not ice-nucleation active at temperatures > -38 °C (Gao et al., 2021) and also other anthropogenic pollution does not contribute INP active at warmer temperatures (Chen et al., 2018). Therefore, the exclusion of periods with touristic activity is not critical for INP observations at nucleation temperatures > -15 °C at Jungfraujoch.

I recommend to specify the time period during which you collected your samples (months, years).

We completed the sentence accordingly:

Here, we analysed INP active at -10 °C (INP-10) and -15 °C (INP-15) collected during FT conditions at the high-altitude observatory Jungfraujoch between January 2019 and March 2021. [page 1, lines 9 and 10]

**Introduction**

Lines 22 – 24: Please check the grammatic of this sentence. I am unsure if I understand the sentence correctly. Do you mean that this community of airborne microorganisms have their emission sources within several tens of kilometers (local/regional scale)?

That is correct. The sentence has been changed accordingly:

A community of airborne microorganisms, for example, is in the PBL mainly composed of organisms emitted from sources within a distance of several tens of kilometers around an observation point (Tignat-Perrier et al., 2019). [page 1, lines 22 to 24]

Line 27: The statement regarding altitude for mixed-phase cloud occurrence is not correct, see my comment before.

In line with our reply to your comment before we have changed the sentence:

**Ice formation in clouds above lowlands in western Europe starts at a few kilometers altitude at around -5 °C (Kanitz et al., 2011). [page 1, lines 27 and 28]**

Line 28: Please add more citations to this very important statement. E.g., many airborne INP measurements are conducted in the free troposphere, and there are more free-tropospheric measurements sites where INP studies were conducted.

Many of these studies have recently been summarised by He et al. (2021), i.e. in the study we refer to. We have made this more explicit by revising the statement and adding another reference:

**However, FT air masses have been little explored in terms of the INP they carry, let alone INP active > -15 °C. For recent summaries of INP studies in the FT see Lacher et al. (2018) and He et al. (2021). [page 1, lines 28 to 30]**

Lines 32 - 34: The cited percentages refer to ice particle residuals, not to the INP concentration measurements at -31°C.

Thank you for pointing out this mistake. Reading the cited references again more carefully and also the papers recommended in your next comment, we have gained additional insights and changed the introduction (see response to next comment).

Lines 35 – 36: I recommend to also include citations from ice particle residual analysis performed at Jungfraujoch (e.g., Mertes et al., 2007; Kupiszewski et al., 2015; Schmidt et al., 2017; Lacher et al., 2021).

Thank you for drawing attention to the ice residual analyses performed at Jungfraujoch. They have prompted us to expand the Introduction:

Yet, in the PBL and at more moderate supercooling (here:  $\geq$  -15 °C), biological particles seem to constitute the majority of the INP population as revealed by heat tests (Hill et al., 2016). Heat deactivates biological INP but leaves mineral INP largely unaffected (Hill et al., 2016). Chen et al. (2021) found during dust events sampled in Beijing that 70% of INP active at  $\geq$  -15 °C (INP-15) were heat-sensitive. Observed heat-sensitive fractions of INP-15 in the PBL of agricultural areas in the USA (Suski et al., 2018) and in Argentina (Testa et al., 2021) were even larger (> 90%). These findings contrast with a small biological fraction detected among ice particle residuals collected at Jungfraujoch from mixed-phase clouds and classified by physico-chemical analyses (Mertes et al., 2007) or by laser ablation mass-spectrometry (Schmidt et al., 2017; Lacher et al., 2021). Not every ice particle residual has initiated as INP the formation of the ice particle it was recovered from, in particular not when it was recovered from a secondary ice particle. An ice particle residual recovered from a primary ice particle and classified by mass spectrometry as mineral dust may have been activated at moderate supercooling by a minor, ice-nucleation active biological component sticking to its surface (Augustin-Bauditz et al., 2016). A heat test, which specifically targets the icenucleation active component, would have classified the same assembly as biological INP. [page 1, line 37 to page 2, line 49]

The recommended paper by Kupiszewski et al. (2016) is concerned with black carbon as a potential INP and would not fit well into the discussion about the fraction of biological INP.

**Material and methods**

I recommend to specify the time period during which you collected your samples (months, years) and to refer to the respective table in the appendix.

**Done.**

Line 52: What is the error associated to a counting by eye?

With good light conditions in the laboratory and an experienced, concentrated observer the error is negligible. Both was the case in our study. Some time ago we have done tests with two observers analysing the same filter independently and found no statistically significant differences between the results.

Lines 71: I recommend to also cite Herrmann et al. (2015) here.

**Done.**

Lines 80 – 81: Please provide more information about the method used to identify impact from Saharan dust event. Which threshold has been used? Has such a method been established in previous work, resp., has it been validated against other measurements or back trajectory calculations (see also my comment above)?

**We hope to have satisfied this demand with our response to your similar comment above.**

Line 82: It is not clear to me why you analyzed the median radon concentration here, while before (line 76) the mean radon concentration was used to identify free tropospheric conditions.

In line 82 we compare the log-normal distributions of hourly radon concentrations as shown in Figure 1. The issue in line 76 are individual hourly measurements of radon concentration. On time scales shorter than one hour, radon may equally be log-normally distributed. However, our detector integrates counts over 30 min. A shorter integration time would make little sense because of the low number of counts observed especially during FT conditions. Therefore, we can only provide an hourly mean, which probably is not that far from what an hourly median based on a shorter integration time would be.

**Results and discussion**

Lines 128 – 135: Your discussion is based on the assumption that during Saharan dust events all aerosol particles are dust particles. With the methods used here you cannot quantify the fraction of dust particles in the overall particle population.

The PM10 mass was 19 times larger on SD filters as compared to FT filters. If we assume that the FT aerosol background is always present and SD mixes with it, then  $18/19^{\text{th}}$  of the aerosol particles (by mass) during SD events are likely dust particles. As shown by Schwikowski et al. (1995), SD predominantly adds to the coarse end (optical diameter > 0.5  $\mu$ m) of the particle volume size distribution, which is the relevant size range for INP (DeMott et al., 2010). During background conditions this end of the distribution is quite flat.

Lines 151 – 153: During Saharan dust events, the total number concentration of aerosol particles does not necessarily increase; more important here would be to quantify the concentration of dust particles and compare it to an increase in INP concentration.

Yes, the total aerosol number concentration may not increase during SD events. However, as mentioned above, the particle volume size distribution increases greatly in the particle size range that is most closely related to INP number concentrations. We have expanded the relevant part of the section accordingly:

This assumption is based on the finding that a large fraction of particle volume (i.e. mass) during a SD event at Jungfraujoch falls into the size range of particles > 0.5  $\mu$ m in optical diameter (Schwikowski et al., 1995), which is a good predictor of atmospheric INP-15 (DeMott et al., 2010; Mignani et al., 2021). [page 5, line 159 to line 161]

Technical

Line 27: I assume you mean "-5 °C".

Yes, thanks.

Line 30: It should read "-31 °C".

Corrected.

Line 45: I assume it should read "At the site,..."?

Yes, thanks.

Line 60: "Sample" instead of "Samples"?

Corrected.

[revised manuscript text omitted]